CellPress

## Commentary

# A standard for sharing spatial transcriptomics data

Kayla C. Jackson[1,2] and Lior Pachter[2,3,*]
[1]Keck School of Medicine, University of Southern California, Los Angeles, CA, USA
[2]Division of Biology and Biological Engineering, California Institute of Technology, Pasadena, CA, USA
[3]Department of Computing and Mathematical Sciences, California Institute of Technology, Pasadena, CA, USA
*Correspondence: lpachter@caltech.edu

Spatial transcriptomic technologies have the potential to reveal critical relationships between the function of genes and cells and their spatial organization. Here, we provide a sharing model for spatial transcriptomics data with the aim of establishing a set of primary data and metadata needed to reproduce analyses and facilitate computational methods development.

## Introduction

Methods for spatial transcriptomic profiling are developing rapidly,[1] with technologies able to capture the expression of hundreds[2,3] to thousands[4,5] of genes at increasingly high resolution. The realization that these developments depend on the dissemination of high-quality data is underscored by policies from major funding agencies that recommend unrestricted release of data.[6,7] These recommendations complement the work of organizations that create standards to ensure persistent access to data[8–10] and journals that require data availability statements in their publications. Unfortunately, meeting these expectations is challenging for researchers working in spatial transcriptomics, as image data can be unwieldy, and there are many intermediate data types that could be beneficial for analysis. As a result, there is no currently accepted standard for spatial transcriptomics data sharing or even agreement on what constitutes sufficient sharing to meet journal and funding agency expectations.

Spatial transcriptomics technologies can be broadly divided into imaging-based and sequencing-based technologies. Imaging-based technologies rely on hybridization of a fluorescently labeled probe that is complementary to mRNA, followed by image-based quantification. These technologies encompass *in situ* hybridization (ISH) and *in situ* sequencing (ISS) methods and capture spatial information over sequential rounds of hybridization and imaging. These methods vary in the number of hybridizations, probes and

gene targets, and instrumentation used to generate the microscopy images. The complexity of the imaging parameters alone has led to development of community guidelines that suggest the minimum metadata required to reproduce an imaging experiment.[11,12] Alternatively, technologies based on next-generation sequencing (NGS) are broadly based on region selection followed by NGS or positional barcoding of regular (e.g., 10X Visium) or irregular arrays (e.g., Slide-seq[13]). Sequencing-based technologies vary in their resolution and can have associated images that represent features that are not captured at the same resolution as the image. Images that require region selection are accompanied by careful annotation of the location within a tissue and the features of the tissue space. These technologies can ostensibly generate coordinates corresponding to specific locations in a histological region, microscopy images, a count matrix, and various metadata, and these disparate forms of data have their own guidelines for dissemination.[14,15]

Here, we provide recommendations for the dissemination of spatial transcriptomic data based on successful models for data sharing in imaging,[11,12,16,17] high-throughput sequencing,[14,18] and single-cell RNA sequencing (scRNA-seq) experiments.[15,19,20] Certainly, previous frameworks have modeled FAIR principles,[10] ensuring findability, accessibility, interoperability, and reusability of experimental data. While researchers cannot anticipate and accommodate all the ways

in which their data will be used, we believe that adherence to a handful of principles we outline can be beneficial. These recommendations are based on existing data types and infrastructure for data storage and transmission but are extensible for larger and more complex datasets. We recognize that even when researchers know what data to share, the question of where to share it is not trivial to answer, and we discuss current solutions to ensure persistent and broad access to their data.

## Sharing images

Images generated from fluorescent microscopy are the basis for several current spatial transcriptomic technologies and, as a standard, should be shared freely. Researchers must decide at which step in the processing pipeline images should be shared. On one hand, sharing raw images offers end users the greatest flexibility for reanalysis and the ability to develop statistical methods. In contrast, sharing pre-processed images means that end users are not required to have expertise in the software and analysis tools for image processing. The tradeoffs for these sharing paradigms are discussed in Table 1.

Both raw and processed microscopy images can feasibly consume gigabytes to terabytes of memory. It is also likely that storage requirements will grow in the coming years. Repositories for hosting image datasets must have both the capacity and flexibility to accommodate growth and potentially more complex images in the future. Spatial transcriptomics

**Table 1. Advantages and disadvantages of image-sharing paradigms**

| Raw images | | Pre-processed images | |
|---|---|---|---|
| Advantages | Disadvantages | Advantages | Disadvantages |
| Allows for the complete reanalysis of the data | Requires image analysis expertise | Does not require image analysis expertise | Images are biased by choices of pre-processing methods made by the authors |
| Can compare upstream image analysis pipeline for different technologies (e.g., segmentation, background reduction, dot detection) | Potentially in proprietary formats; cross-platform incompatibility | Fewer analytical demands | May not be compatible with other datasets |
| Allow data from multiple studies to be processed uniformly | Requires extensive storage | Useful for end users focusing on re-analysis of downstream results | Requires extensive storage; but potentially less than raw images if many images are stitched together |
| Can improve reproducibility — others can verify that the results have not been "cherry-picked" | May not be easy to import into common analysis software | Will not get different results from original authors because of minor differences in pre-processing | Potential for wrong "version" to be shared. Authors may not share processed images that generated results in paper |
| Allow analysis of features that may be called as "background" | May require conversion to other formats before they can be uploaded to public repositories | Common starting point when discussing differences in downstream results | – |
| End-user can experiment with less conservative processing methods | Lack of standardized metadata may make usability challenging | – | – |

datasets published by prominent consortia and private companies exhibit sizes spanning from hundreds of gigabytes to a few terabytes, offering insight into the typical scale of datasets. Several of the broad-spectrum repositories noted in Table 2 can accommodate datasets on this scale and would probably serve the purposes of most researchers using current technologies.

The community guidelines on sharing and reporting imaging data[11] have underscored the importance of metadata standardization with respect to data reuse, and many of these recommendations are relevant to spatial transcriptomics experiments (Table S1). Efforts have been made to encourage use of the open microscopy environment (OME)-TIFF file format,[16,21] and tools[22] have been developed to convert between proprietary image and OME file types. The OME-TIFF file format is near ubiquitous in the bioimaging community and enjoys robust support in bioimaging software packages. Though OME-TIFF can support multi-dimensional data, it is most performant on 2D TIFF tiles and is not optimized for cloud-based storage infrastructure. The OME next-generation file format (NGFF) is emerging as a scalable image format to address these

limitations. Importantly, the OME-NGFF[23] is necessarily extensible and will flexibly support multimodal and multidimensional images in the future. These efforts are arguably necessary to maximize the utility of spatial transcriptomics data.

**Sharing segmentation data**

Image segmentation involves annotating images to distinguish relevant objects from background. Segmentation techniques employ techniques that include random forests,[24] convolutional neural networks,[25] or manual outlining[26] to define the boundaries of regions that contain transcripts. In many contemporary methods, segmentation data fundamentally enable single-cell analysis and encode the spatial context of the tissue under study. This information holds significant utility, positioning it as a crucial component within the sharing paradigm.

Segmentation data are typically conveyed as an image mask where pixels associated with objects and background are populated with distinct values. In the binary case, the mask assigns a value of 0 to pixels that are considered background and a non-zero value (usually 1) to pixels considered part of a feature in the image. Researchers can use standard image pro-

cessing software like ImageJ[26] to extract the coordinates for each object. Notably, segmentation masks are sensitive to the specific tools used to process the input image. In cases where segmentation data are not shared, end users looking to reproduce segmentation results may be unable to deduce heuristic choices made by the original researcher. Sharing segmentation data either as an image mask or as a series of coordinates easily circumvents this challenge. Additionally, since segmentation is not typically performed on every image, researchers should unambiguously indicate which images were used as input to segmentation algorithms.

The utility of the segmentation mask is diminished without identifiers that relate the segmented objects to other analysis data. In the segmentation mask, pixels associated with cells can be assigned a value corresponding to a unique cell identifier. Alternatively, researchers can supply object coordinates in a tabular format and trivially add a column for an identifier. Many NGS-based methods use spatially indexed sequencing barcodes rather than image segmentation to encode spatial context. Here, a spatial barcode represents a "spot" on a microscopy slide and introduces the possibility that sequenced

**Table 2. Storage capacities and fees associated with image data repositories**

| Host | Scope | Individual file size limit (GB) | Dataset size limit | Additional storage cost and fees | Closed-access options? | Comment |
|---|---|---|---|---|---|---|
| Figshare | Broad spectrum | 5 | 20 GB for free accounts; 5 TB for paid accounts | $585 per 250 GB; $160 processing fee | Yes | – |
| Zenodo | Broad spectrum | None stated | 50 GB for free accounts; larger datasets accepted on case-by-case basis | Accepts donations | Yes | – |
| Dryad | Broad spectrum | 10 | 300 GB | $50 per 10 GB; $120 Publishing charge | No | PPI is not allowed |
| Image Data Resource | Cells, tissue | None stated | None stated | None stated | Limited | Contact if dataset >10TB |
| Cell Image Library | Organisms, cell types, and cellular processes | None stated | None stated | None stated | No | – |
| BioImage Archive | Bioimaging data from molecule to organism scale | None stated | None stated | None stated | No | File upload recommendations change based on size of project |
| EMPIAR | Electron microscopy | 1,000 | None stated | None stated | No | Only justifiable changes are permitted after release |
| GigaDB | Broad spectrum | None stated | None stated | None stated | No | Only hosts data from manuscripts accepted in Gigascience |
| Kaggle | Broad spectrum | 2 | 100 GB | None stated | Yes | – |
| ODAP/ Harvard DataVerse | Broad spectrum | 2.5 | 1 TB | None stated | Yes | Contact for datasets >1TB |
| IEEE Data Portal | Broad spectrum | None stated | 2 TB; Institutional accounts have access up to 10 TB/dataset | $1950 open access fee | Limited to paid accounts | – |

barcodes overlap with several segmented features. Researchers should provide a mapping file that describes how these spatial encodings intersect (Table S1), particularly in cases where a spatial barcode captures more than one segmented object.

## Sharing sequencing data

Many journals require that sequencing data be deposited in repositories that adhere to MIAME/MINSEQE guidelines[14,18] or that are part of the International Nucleotide Sequence Collaboration (INSDC).[27] Furthermore, efforts to describe additional metadata that should be shared with scRNA-seq experiments[15] are relevant to spatial transcriptomics experiments (Table S1). Among the most popular repositories are the EMBL-EBI ArrayExpress,[28] European Genome Phenome Archive (EGA),[29] NCBI Gene Expression Omnibus (GEO),[30] and NCBI Database for Genotypes and Phenotypes (dbGaP).[31] The GEO and ArrayExpress provide unrestricted access, while the dbGaP and EGA are suitable for researchers that require controlled access to data containing protected information. These resources all provide archival storage along with an identifier that facilitates later access. Repositories that are part of INSDC are committed to FAIR principles and are in support of a framework that ensures that sequencing data are permanently available to the scientific community.

## Sharing expression matrices and metadata

The expression matrix serves as the fundamental analytical unit for almost all spatial transcriptomic analyses. It represents the raw or processed count of the number of RNA molecules corresponding to each gene detected in each cell. While sharing the count matrix alone obscures the upstream processing steps involved in its generation, their portability in a variety of formats makes them practical to share. The unprocessed count matrix, prior to any filtering or normalization, and raw sequencing data would provide the greatest benefit to end users. Nevertheless, it remains important for researchers to share a thorough description of the downstream processing steps that were employed to generate the matrix. Importantly, the identifiers in the count matrix should be linked to those submitted to sequencing and imaging repositories. Models have been proposed for the types of metadata that should be shared with count matrices generated from scRNA-seq technologies,[15] but complex metadata for spatial experiments are not easily transferable and can require custom infrastructure for their conveyance.

Two examples of metadata with special relevance include the geometries or shapes of segmented objects and the locations of individual transcripts, with the latter being available in technologies that use ISH for expression profiling. The geometry information is typically conveyed as coordinates, in pixel or physical space, defining the boundaries and center of the object. The coordinates of transcripts are typically located within these boundaries. Sharing these data in a memory-efficient manner that supports indexing across the dimensions of the expression matrix can be challenging, especially as the number of cells and genes profiled in a single experiment continues to grow. Various representations, such as SpatialFeatureExperiment[32] in R, have been proposed, drawing inspiration from geospatial data analysis and integrating standards for storage and access of spatial geometries into objects familiar to biologists. By utilizing methods developed for geospatial analysis, biologists can take advantage of existing tools to explore spatial relationships. Ongoing efforts are focused on adapting these methods to take advantage of on-disk op-

erations and on-demand memory loading to accommodate large datasets. The new SpatialData[33] library is based on the OME-NGFF framework and addresses many of these memory challenges, but its adoption requires end users to have familiarity with Python. In keeping with FAIR principles, the ideal implementation should support interoperability across platforms and embed metadata alongside the count matrix.

## Conclusion

The issue of ensuring FAIR access to data is not unique to the spatial transcriptomics field and has been considered elsewhere,[34] but the challenges associated with sharing a combination of imaging, sequencing, and tabular data demand particular attention. The ideal data sharing paradigm should balance the difficulty imparted on the researchers generating the data and the utility to end users.

We have developed recommendations for sharing of spatial transcriptomics data based on the data types produced by existing technologies and storage capacities of widely used data repositories. However, spatial transcriptomics is a rapidly evolving field and advances in sequencing and imaging technologies will necessarily require that sharing standards be updated to accommodate new components that are not considered here. For example, efforts to combine spatial transcriptomics with three-dimensional imaging techniques will reasonably lead to the generation of multidimensional expression matrices and annotations. The exact amount of additional memory that will be required solely for storing data generated by these advancements poses a challenge. It is reasonable to anticipate that previous estimates of a typical dataset size ranging from several hundred gigabytes to a few terabytes may soon become outdated. Addressing these challenges will require improved infrastructure and the utilization of cloud-based solutions. We also anticipate that overcoming these challenges will emphasize collaboration and open data-sharing initiatives to support further progress.

The increasing volume of spatial transcriptomics datasets combined with the complexity of human sequencing datasets underscores the need to implement robust restrictions to safeguard genetic privacy.

Achieving a balance between data utility and privacy protection can be accomplished by reinforcing data access controls and establishing comprehensive data use agreements. Additionally, as more data are being migrated to cloud-based servers, it is crucial to regularly monitor repositories containing human data to ensure secure storage protocols are in place both during transit and archiving. Furthermore, as policy changes are adapted to include evolving technologies and emerging privacy risks, it is important to develop protocols that will retroactively and automatically secure existing datasets to align with policy changes. These considerations are essential for facilitating scientific advancements and retaining the trust of research participants.

The challenges faced by spatial transcriptomics researchers cannot be understated, but, given that scientific progress rests on the ability of researchers to use existing data to pose questions not yet considered, the difficulties in sharing data must be balanced against the needs for transparency, reproducibility, and usability. We encourage researchers to examine their goals for data sharing at each step of the analysis pipeline. By taking these considerations into account, the broader scientific community can fulfill its promise to ensure that research data are accessible and that the results derived from them are reproducible.

### SUPPLEMENTAL INFORMATION

### DECLARATION OF INTERESTS

The authors declare no competing interests.

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
