## [Document S1. Transparent peer review records for Jackson et al · Cell Genomics]

A standard for sharing spatial transcriptomics data

Kayla C. Jackson^{1,2}, Lior Pachter^{2,3}

Summary

Initial submission: Received : 12/1/2022

Scientific editor: Laura Zahn

First round of review: Number of reviewers: 2
Revision invited : 3/10/2023
Revision received : 5/4/2023

Second round of review: Number of reviewers: 0
Accepted : 7/7/2023

Data freely available: N/A

Code freely available: N/A

This transparent peer review record is not systematically proofread, type-set, or edited. Special characters, formatting, and equations may fail to render properly. Standard procedural text within the editor's letters has been deleted for the sake of brevity, but all official correspondence specific to the manuscript has been preserved.

Referees' reports, first round of review

Reviewer #1: This commentary addresses an important issue of spatial transcriptomics (ST) data sharing. Many questions during ST data sharing have been discussed. I have only one minor suggestion, i.e., whether the authors prospectively propose a universal count matrix format for the currently available and potentially future ST platforms to share the processed count matrices. This is important because the formats of inputs are fundamental to downstream algorithm development. If a universal format is available, it will save large amount of work and time for developers devoted to mining the biomedical values hidden in ST data from different platforms. Successful examples are fasta, sam, bam, and gff files for Sanger and NGS data. ST data need new formats.

Reviewer #2: Apologies for our delayed reply and thank you for requesting our review on the paper entitled: "A standard for sharing spatial transcriptomics data"

This paper is interesting and timely considering the increase in popularity of spatial transcriptomics datasets being submitted to public resources. An understanding of the technical and biological requirements will help both data owners and interested partners handling this data type to accurately represent it and promote the reuse and reproducibility of these data.

The authors clearly understand the importance of including the imaging information alongside the sequencing data with this type of data allowing the overlay of clustered expression profiles to the original tissue sample. An overview of existing microscopy repositories is helpful, as shown in Table 1 although some justification of why certain formats are considered of higher standards than others is missing, which readers may find helpful.

Unfortunately, given the title and intended topic, we do not feel that there is sufficient depth or information provided regarding the spatial data itself or the integration between sequencing data and imaging data. We would therefore recommend the following:

Major concerns:

1. Within the introduction section, we would like to see a broader introduction to spatial data. Currently what is covered is quite superficial with no consideration of different library techniques in technical detail only a brief consideration of different technologies which isn't extensive - smFISH and spatial barcoding only mentioned. A broader introduction that gives more details of how spatial sequencing is set up would be beneficial to the reader and confer a greater understanding as to why specific elements need to be recorded and open up the questions around how to collect and represent these.
2. In the 'Sharing images' section, there is no consideration for requirements describing microscopy metadata or hardware description which is also a consideration for reproducible analysis. How a sample is treated for imaging and captured have major recognized implications for the downstream analysis and is currently being addressed in detail by the community.
3. We find it of considerable concern in the 'Sharing expression matrices and metadata' section that the authors claim 'spatial data does not lend itself to tabular format' or that 'count matrices' as a format for data sharing is not easily transferable. As we are currently capturing this in tabular format for both meta and technical data at both the sample and single entity (both spot and cell) level this seems unconsidered. Upstream processing of count matrices need to also be shared, a point which is bypassed by the authors. Additionally, the AnnData format described here: <https://doi.org/10.1101/2021.12.16.473007> is becoming more widely accepted as standardized format for single cell expression matrices combining metadata with expression count matrices in an easily transferable and reusable format.

Minor concerns:

1. Although, briefly referred to in the introduction and 'Sharing images' sections, we would like to see greater mention of community guidelines for microscopy (REMBI guidelines detailed here: <https://doi.org/10.1038/s41592-021-01166-8>) and greater consideration of the technical elements of a single cell sequencing dataset (MINSCE, although the paper itself was cited). We believe that highlighting their role in describing how these data are captured and described consistently is of great importance and increases their scope and uptake by the scientific community.
2. The authors give recommendations for image repositories in Table 1 and the 'Sharing images' sections but fail to consider guidelines for linking the images back to sample metadata or sequencing files - major requirements for data reuse.
3. We would ask the authors to consider the inclusion of a similarly in-depth recommendation of sequencing databases to complement the microscopy resources in Table 1. How are different archives implementing FAIR principles and metadata standards at both the biological and technical level.

Authors' response to the first round of review

Responses to comments made by reviewer 1

Minor comment 1:

I have only one minor suggestion, i.e., whether the authors prospectively propose a universal count matrix format for the currently available and potentially future ST platforms to share the processed count matrices.

Response:

Thank you for your comments. We agree that a standard format for conveyance of ST data would improve facilitate research in the field. Our lab is actively developing infrastructure for this purpose (see SpatialFeatureExperiment [10.18129/B9.bioc.SpatialFeatureExperiment]) in both R and Python programming languages facilitate interoperability between them. We believe this parallel approach motivates compatibility and supports technology-agnostic development.

Responses to comments made by reviewer 2

Additional comment:

An overview of existing microscopy repositories is helpful, as shown in Table 1 although some justification of why certain formats are considered of higher standards than others is missing, which readers may find helpful.

Response:

Thanks- we agree that inclusion of a discussion of why researchers may favor one imaging repository over another is useful. Our recommendations are broad. Researchers should consider the repositories that meet their storage needs and balance long term costs, which we recognize can be significant in some cases. We have observed in the spatial transcriptomics literature that researchers have indicated that the storage requirements for imaging data make it infeasible to share. The table in the manuscript is included to highlight that there are repositories that can reasonably accommodate imaging data. Rather than recommend a specific repository, we have updated the manuscript to restate the call made by the authors of the REMBI, that all authors who produce imaging data share it in an appropriate database or repository.

Comment 1:

Within the introduction section, we would like to see a broader introduction to spatial data. Currently what is covered is quite superficial with no consideration of different library techniques in technical detail only a brief consideration of different technologies which isn't extensive - smFISH and spatial barcoding only mentioned. A broader introduction that gives more details of how spatial sequencing is set up would be beneficial to the reader and confer a greater understanding as to why specific elements need to be recorded and open up the questions around how to collect and represent these.

Response 1:

We have updated the introduction section to include an overview of the technical details of various spatial technologies. We have broadened the description of these methods and rely heavily on reviews of existing spatial methods that describe these technologies in detail as we are limited by the word limit of this article format.

Comment 2:

In the 'Sharing images' section, there is no consideration for requirements describing microscopy metadata or hardware description which is also a consideration for reproducible analysis. How a sample is treated for imaging and captured have major recognized implications for the downstream analysis and is currently being addressed in detail by the community.

Response 2:

We agree that there should be requirements for describing microscopy metadata and hardware; omission of this was an oversight on our part. We have added a Supplementary Table 1 (and a reference to it) that enumerates sharing recommendations, including those for microscopy metadata, for spatial transcriptomics experiments. This is in keeping with similar frameworks that have been published for other experimental methods.

Comment 3:

We find it of considerable concern in the 'Sharing expression matrices and metadata' section that the authors claim 'spatial data does not lend itself to tabular format' or that 'count matrices' as a format for data sharing is not easily transferable. As we are currently capturing this in tabular format for both meta and technical data at both the sample and single entity (both spot and cell) level this seems unconsidered. Upstream processing of count matrices need to also be shared, a point which is bypassed by the authors.

Additionally, the AnnData format described here:

<https://doi.org/10.1101/2021.12.16.473007> is becoming more widely accepted as standardized format for single cell expression matrices combining metadata with expression count matrices in an easily transferable and reusable format.

Response 3:

Thank you for your feedback on this section. You are right that the original language in the text may not have clearly conveyed the intended point. The quote from the manuscript that "[spatial] data does not lend itself easily to a tabular format" was intended to refer specifically to the geometries of segmented cells and spot locations of individual transcripts detected by FISH methods. While tabular formats are well-suited to share metrics (e.g. area, eccentricity) derived from the geometry data, subsequent data users lose the option to compute additional metrics or perform geometric operations in future analysis without access to the original geometries, and our intent was to point out that shortcoming. One solution for this is to simply share a segmentation mask, and we certainly contend that this should be done (Sharing segmentation data). However, if authors fail to provide adequate identifiers that link segmented objects to the count matrix and celllevel metadata, then this is useless. Another option would be to share a representation of the geometric object in the Cartesian plane. While tools do exist to represent these objects in a data frame-like object (e.g. GeoPandas in Python, sf in R) these objects can consume a considerable amount of memory, especially as the number of coordinates required to sufficiently represent the geometry increases. Furthermore, the spot locations for individual transcripts in a cell are an inherently jagged array as different numbers of transcripts are detected in each cell. Storing jagged metadata in the experimental object is currently being explored in packages like AnnData, but these features are still experimental (<https://anndata-tutorials.readthedocs.io/en/latest/awkwardarrays.html>). These data can certainly be contained in separate objects, but in an ideal case, the experimental data and metadata can be stored together so that all components are linked across representations and analysis tasks. We've updated the manuscript to better explain these nuances related to sharing of count matrices, or count matrices only.

Minor comment 1:

Although, briefly referred to in the introduction and 'Sharing images' sections, we would like to see greater mention of community guidelines for microscopy (REMBI guidelines detailed here: <https://doi.org/10.1038/s41592-021-01166-8>) and greater consideration of the technical elements of a single cell sequencing dataset (MINSCE, although the paper

itself was cited). We believe that highlighting their role in describing how these data are captured and described consistently is of great importance and increases their scope and uptake by the scientific community.

Response to minor comment 1:

The community guidelines should certainly be highlighted in the text. We have added additional mentions of these guidelines in the relevant sections of the manuscript and in Supplementary Table 1.

Minor comment 2:

The authors give recommendations for image repositories in Table 1 and the 'Sharing images' sections but fail to consider guidelines for linking the images back to sample metadata or sequencing files - major requirements for data reuse.

Response to minor comment 2:

We have updated the text to reflect the importance of providing identifiers that link observations across sequencing, imaging, and metadata files and have also added this recommendation to Supplementary Table 1.

Minor comment 3:

We would ask the authors to consider the inclusion of a similarly in-depth recommendation of sequencing databases to complement the microscopy resources in Table 1. How are different archives implementing FAIR principles and metadata standards at both the biological and technical level.

Response to minor comment 3:

We considered adding such a table but almost all sequence data is deposited in GEO/dbGAP/SRA and these are compliant with MIAME/MINSEQE. Rather than adding a table we've added some remarks about this in the manuscript; we agree that it's useful to include this given that a lot of spatial analysis happens in the context of, or with, sequencing data.